# Levels of Stress, Anxiety, and Depression in University Students from Spain and Costa Rica during Periods of Confinement and Virtual Learning

**José Gijón Puerta [1]**, **María Carmen Galván Malagón [2]**, **Meriem Khaled Gijón [3]** and **Emilio Jesús Lizarte Simón [1,*]**

[1] Department of Didactics and School Organization, University of Granada, 18071 Granada, Spain
[2] Department of English Philology, University of Extremadura, 06006 Badajoz, Spain
[3] Laboratory for Cognition, Health, Training and Interaction among Humans, Animals and Machines, University of Granada, 18071 Granada, Spain
[*] Correspondence: elizarte@ugr.es

**Abstract:** Mental health problems, specifically those related to stress, anxiety, and depression, have become more prevalent among college students compared to data available prior to the levels of the COVID-19 pandemic. Recent studies have shown that in different geographical areas, there is a high prevalence of depression and anxiety symptoms in university students compared to pre-pandemic levels. Thus, our objective was to establish self-perceived levels of stress, anxiety, and depression in university students earning an education degree at the University of Granada and the University of Costa Rica during periods of confinement and virtual learning associated with the COVID-19 pandemic. The final study sample consisted of 942 students from both universities. Two questionnaires were administered: The state trait anxiety inventory and the depression, anxiety, and stress scale 21. Descriptive analyses, mean comparisons, Pearson correlation coefficients, and multivariate regression were performed. Reliability was assessed using Cronbach's alpha, and the effect size was analyzed using Cohen's *d*. The results indicated that levels of depression, anxiety, and stress were mild or moderate despite the confinement and virtual learning associated with the COVID-19 pandemic. Women had higher levels of anxiety than men, and singles had higher levels of anxiety than individuals in other family situations. Younger individuals had higher levels of stress and anxiety.

**Keywords:** higher education; anxiety; depression; college students; stress; mental health





## 1. Introduction

The health crisis resulting from COVID-19 pandemic is a turning point, from which new teaching scenarios are being generated, and their consequences on the mental health of students should be investigated. Mental health problems have become a more frequent phenomenon among university students [1], especially those related to stress, anxiety, and depression [1,2]. In the United States, the college-age population has been described as the group with the highest level of stress [3]. Before the pandemic, 37.4% of students who requested psychological support had relevant problems, mainly anxiety and depression [4]. Some studies have highlighted that a significant percentage of first-year students (33% to 40%) have moderate to severe anxiety [5].

As indicated by Ruiz [6] (pp. 97–98), there is a question regarding the differentiation of anxiety and depression symptoms that complicates differential assessments. For example, anxiety and depression show high comorbidity rates, with the most comorbid disorders being anxiety, depression, and panic disorders. Additionally, instruments designed to analyze and measure symptoms of depression and anxiety are strongly correlated with



each other. There is also evidence of a significant prevalence of anxiety and depression disorders in university students that affects academic performance and dropping out [7].

Baranza [8] (p. 274) describes three indicators of systemic imbalance caused by academic stress: physical, psychological, and behavioral. Among the physical indicators are an increase in drowsiness and time spent sleeping, as well as insomnia and chronic fatigue, migraines, tremors, and obsessive reflexes, such as nail biting. Psychological factors usually manifest as blocks, concentration and memory problems, restlessness, anxiety, and depression. Finally, behavioral indicators include isolation, academic absenteeism, reluctance to eat, refusal to perform school-related tasks, never-ending conversations with close people, and increased eating.

In the education field, academic stress is understood as a systemic process that, psychologically and adaptively, occurs in three phases: initial, that is, a person suffers a series of demands they consider "stressors"; intermediate, that is, the "stressors" generate an imbalance that manifests as a series of indicators or symptoms; final, that is, the imbalance impels acts to restore equilibrium in the system [8].

According to [9] (p. 28), depression presents in two ways: as a syndrome with a set of symptoms that defines a pathological state of sadness or bad mood, defined from a criterion of normality, and as a state of emotional pain, sadness, or unhappiness that is produced as a reaction to unpleasant situations. Depression can be approached as a varied disorder that reduces social, interpersonal, and work functioning [10]. However, "stress is linked more to the physiology of the organism, while anxiety is linked to the physiology and psychology of the individual. Anxiety results from stress; stress causes, although not always, anxiety" [11] (p. 18). Anxiety is conceived as an emotional reaction to a threat from a cognitive, physiological, motor, and emotional point of view [12] going through a phase of psychobiological adaptation to a situation of current or future danger [13]. Anxiety disorders can be considered risk indicators for the onset and development of depression [14].

*Depression, Anxiety, and Stress in University Students*

At the international level, studies, such as those by Hunt and Eisenberg [15] in the United States and the United Kingdom, have shown that in recent decades, there has been a greater demand from university students for psychological services, in addition to a greater incidence of more complex emotional disorders that counselling teams must alleviate. In relation to these facts, a university represents a period of change, during which students encounter new experiences and develop new skills, as well as adapting to social and academic life, which facilitates the development of academic persistence [16]. Notably, student retention is one of the most substantial challenges of higher education at the international level, as it has detrimental consequences for students, institutions, and society [16].

For many students, beginning university studies can be a stressful transition, due to changes in relationships between their classmates and the realities of a new way of life that require changes in their personal and academic goals [17,18].

Given that the transition to university involves a high risk of maladaptive coping, the appearance of psychopathology, and academic failure, universities must assume and offer prevention and intervention measures within an integrated mental health care system for students [19].

According to Beiter [20], some of the most relevant concerns of U.S. students are pressures related to academic success and academic performance during their university studies. Thus, the evaluative processes involved in the subjective experience of stress in the university environment are influenced by contextual and psychosocial factors [21].

In Spain, there have been more requests from students for psychological care services because of the new virtual learning model, long-term uncertainty, and changes in the scholarship system, among other concerns [22]. In general, one in five Spaniards shows symptoms of anxiety (19.6%), depression (22.1%), or stress (19.7%), but young people

between 18 and 24 years of age exhibit more indicators of these symptoms [23]. In studies carried out in higher education centers in Costa Rica between 2014 and 2018, there was an increase of 49% in requests for psychological care services [24].

However, it has been observed that anxiety levels decrease with age because of an increased ability to regulate emotions [25,26]. The results of a study by Fernández [27] in a Spanish university indicated that exams, time management, study load, lack of free time, and concern about not meeting the expectations of parents generate stress, anxiety, and depression. Similar issues have been highlighted in higher education students in Costa Rica, as in the study by Ramírez [28], in relation to emotional management in students as a cause of increased stress, and in that carried out by Belhumeur [29], in which academic and financial aspects were the most important sources of stress.

Reviews, such as those by Mac [30], show that in very different geographical areas (United States, Switzerland, and England), there has been a high prevalence of depression and anxiety symptoms in university students, indicating higher levels of anxiety and depression symptoms compared to prior to the COVID-19 pandemic. As such, we proposed the following research objective: to establish self-perceived levels of stress, anxiety, and depression in university students earning an education degree at the University of Granada and the University of Costa Rica during periods of confinement and virtual learning associated with the COVID-19 pandemic. The aim was to determine whether there are significant differences in variables, such as gender, age, country, and pet ownership, which have been studied in recent decades from international perspectives [31–33].

## 2. Materials and Methods

The final study sample consisted of 942 students, 82.6% of whom were women and 17.4% of whom were men, aged between 18 and 58 years, with an average age of 23.8 years (SD = 7). Table 1 provides the demographic data of the participants: 83.4% were single, 73.2% were studying without working, 64.7% were students at the University of Granada, and 35.3% were students at the University of Costa Rica. In total, 68.6% of the participants claimed to have a pet.

**Table 1.** Descriptive demographic variables.

|  | *n* | *%* |
|---|---|---|
| **Civil status** | | |
| Married/living with a partner | 142 | 15.1 |
| Single | 786 | 83.4 |
| Widowed/divorced | 14 | 1.5 |
| **Situation** | | |
| Study | 690 | 73.2 |
| Work and study | 252 | 26.8 |
| **University** | | |
| Granada | 608 | 64.7 |
| Costa Rica | 332 | 35.3 |
| **Pet** | | |
| No | 296 | 31.4 |
| Yes | 646 | 68.6 |

### 2.1. Data Collection

In this study, we used two questionnaires to obtain data on participants' self-perceptions regarding anxiety, stress, and depression.

The state trait anxiety inventory (STAI) questionnaire by Spielberger [34] is composed of two subscales with 20 items each that evaluate anxiety as a state (emotional state "at this moment" as a transitory condition) and anxiety as a trait (emotional state "in general" as a stable propensity). Responses are provided using a 3-option Likert scale from 0 to 3, in which 0 is "almost never" and 3 is "almost always". The total score was obtained by adding the scores for each item after converting the reverse-scored items. The total score

ranged from 0 to 60, with a higher score indicating a higher degree of anxiety. The internal consistency coefficients for the state subscale ranged from 0.89 to 0.95, and those for the trait subscale ranged from 0.82 to 0.91 [34].

The depression, anxiety, and stress scale DASS-21 [35] was used to evaluate levels of stress, anxiety, and depression, and to obtain a general mental health picture. The scale is organized into three subscales (depression, anxiety, and stress), with seven items each.

The DASS-21 has 21 items, with four alternative responses in Likert format, ranging from 0 ("It doesn't describe anything that happened to me or felt during the week") to 3 ("Yes, this happened a lot, or almost always"). In order to answer, the question asks the respondent to indicate to what extent the statement describes what happened or felt the person during the last week. This instrument has the advantage of being a self-reported scale, short, easy to administer and answer, and easy to interpret.

The version developed by Fonseca [36] who validated the scale in Spanish university students, was used.

### 2.2. Procedure

The instruments were administered in the 2019/20 academic year, during the period of confinement due to the COVID-19 pandemic, specifically in May 2020, to students at the University of Granada and the University of Costa Rica earning teaching degrees through the Faculty of Education Sciences of both universities. The instruments were administered in online format, during class time. For this purpose, the University of Granda's own platform, the PRADO platform (Plataforma de Recursos de Apoyo a la Docencia), was used. Both the administration of the instruments and the statistical treatment of the data were conducted in accordance with the recommendations of the ethical standards of the University of Granada. The average time to complete both questionnaires was approximately 45 min.

### 3. Results

*Data Analysis*

In the descriptive statistical analysis, for the qualitative variables, the number of cases present in each category and the corresponding percentage were obtained, and for the quantitative variables, the minimum and maximum values and the mean and standard deviation were calculated.

For the comparison of means between two groups, Student's *t*-test was used once the assumptions of normality and homogeneity of variances were verified with the Kolmogorov-Smirnov test and Levene's test, respectively. The Pearson linear correlation coefficient (*r*) was calculated to study the possible relationship between the two variables. Reliability was assessed using Cronbach's alpha and the effect size was assessed using Cohen's *d*.

Multiple linear regression was used to determine the possible effects of demographic variables, the STAI, and scores on the depression and anxiety dimensions of the DASS-21, verifying the model hypotheses through analysis of the residuals.

Statistical analysis was performed using SPSS 25.0 for Windows. Statistical significance was set at $p < 0.05$.

Table 2 shows the means and standard deviations, Cronbach's alpha reliability indices, and correlations between scales. Regarding internal consistency, alpha index values were greater than 0.75, indicating high reliability.

Table 2 shows the STAI and DASS-21 scores, which indicated mostly direct and moderately significant correlations between all the factors whose factors were between $r = -0.099$ ($p < 0.05$) and $r = 0.748$ ($p < 0.001$). More specifically, we observed a significant direct correlation between state anxiety and trait anxiety ($r = 0.259$, $p < 0.001$) and a significant indirect correlation between state anxiety and depression ($r = -0.099$, $p < 0.05$). Thus, the higher a student's state anxiety level, the higher their level of trait anxiety, and the higher a student's state anxiety level, the lower their level of depression. Regarding trait anxiety, there were significant direct correlations with depression ($r = 0.463$, $p < 0.001$),

anxiety ($r = 0.464$, $p < 0.001$), and stress ($r = 0.415$, $p < 0.001$). Therefore, the higher a student's state anxiety, the higher their levels of depression, anxiety, and stress. Regarding depression, there were significant direct correlations between depression and anxiety ($r = 0.676$, $p < 0.001$) and between depression and stress ($r = 0.748$, $p < 0.001$). Therefore, the higher a student's level of depression, the higher their levels of anxiety and stress. Finally, there was a significant direct correlation between anxiety and stress (r = 0.680, $p < 0.001$). Thus, the higher a student's anxiety level, the higher their level of stress.

**Table 2.** *Mean* (SD), reliability and correlations between scales.

|  | *Mean* (SD) | Cronbach's Alpha | 1 | 2 | 3 | 4 |
|---|---|---|---|---|---|---|
| 1. STAI state | 28.66 (4.46) | 0.824 | 1 |  |  |  |
| 2. STAI trait | 28.08 (5.69) | 0.820 | 0.259 *** | 1 |  |  |
| 3. DASS depression | 5.54 (4.88) | 0.790 | −0.099 * | 0.463 *** | 1 |  |
| 4. DASS anxiety | 4.29 (3.95) | 0.812 | 0.057 | 0.464 *** | 0.676 *** | 1 |
| 5. DASS stress | 7.55 (4.33) | 0.801 | −0.023 | 0.415 *** | 0.748 *** | 0.680 *** |

\* $p < 0.05$; \*\*\* $p < 0.001$.

When observing the average scores evaluated in this study with the two instruments used, we can see that the STAI state scale had a mean of 28.66 (SD = 4.46), corresponding to the 75th percentile, and a Cronbach's alpha of 0.824; the STAI trait scale had a mean of 28.08 (SD = 5.69), corresponding to the 70th percentile, and a Cronbach's alpha of 0.820. On the DASS-21, the averages were consistent with mild depression ($M = 5.54$; SD = 4.88), with a Cronbach's alpha of 0.790, mild anxiety ($M = 4.29$; SD = 3.95), with a Cronbach's alpha of 4.29, and mild stress ($M = 7.55$; SD = 4.33), with a Cronbach's alpha of 0.801.

Table 3 shows the descriptive scores for the scales based on gender, as well as the results of Student's *t*-tests performed to determine if there were statistically significant differences between men and women. There were statistically significant differences between men and women in the anxiety dimension, that is, the score for men ($M = 3.23$, DT = 3.52) was significantly lower than that for women ($M = 4.51$, DT = 4.01). No differences were observed between men and women in other dimensions.

**Table 3.** *Mean* (SD), reliability and correlations between scales.

|  | Gender, *Mean* (SD) | | Student's *t*-Test | | *d* |
|---|---|---|---|---|---|
|  | **Male** | **Female** | *t* **(469)** | *p***-Value** |  |
| STAI state | 29.46 (4.43) | 28.49 (4.46) | 1.793 | 0.074 | 0.22 |
| STAI trait | 27.27 (4.94) | 28.25 (5.82) | −1.425 | 0.154 | −0.17 |
| DASS depression | 5.72 (5.08) | 5.50 (4.84) | 0.372 | 0.71 | 0.05 |
| DASS anxiety | 3.23 (3.52) | 4.51 (4.01) | −2.677 | 0.008 | −0.33 |
| DASS stress | 6.90 (4.63) | 7.69 (4.26) | −1.502 | 0.133 | −0.18 |

SD: Standard deviation; *d*: Cohen's effect size.

Table 4 shows the descriptive scores for the scales based on relationship status, as well as the results of Student's *t*-tests performed to determine if there were statistically significant differences between individuals who were single and those who had another relationship status. There were significant differences between groups in the depression dimension, that is, the score for single students ($M = 5.86$, DT = 4.89) was significantly higher than that for individuals with any other relationship status ($M = 3.91$, DT = 4.49). There was also a statistically significant difference between groups in the stress dimension, that is, the score for singles ($M = 7.79$, DT = 4.25) was significantly higher than that for individuals with any other relationship status ($M = 6.36$, DT = 4.55).

**Table 4.** Descriptive and comparative scores for the scales based on relationship status.

| | Relationship Status, *Mean* (SD) | | Student's *t*-Test | | *d* |
|---|---|---|---|---|---|
| | **Single** | **Other** | ***t* (469)** | ***p*-Value** | |
| STAI state | 28.65 (4.45) | 28.71 (4.56) | 0.092 | 0.926 | 0.01 |
| STAI trait | 28.27 (5.70) | 27.10 (5.57) | −1.666 | 0.096 | −0.21 |
| DASS depression | 5.86 (4.89) | 3.91 (4.49) | −3.257 | 0.001 | −0.40 |
| DASS anxiety | 4.43 (3.91) | 3.55 (4.10) | −1.803 | 0.072 | −0.22 |
| DASS stress | 7.79 (4.25) | 6.36 (4.55) | −2.687 | 0.007 | −0.33 |

SD: Standard deviation; *d*: Cohen's effect size.

To determine whether there was a relationship between age and scores on the dimensions, the Pearson correlation coefficient (*r*) was calculated. As shown in Table 5, there was a statistically significant indirect relationship between age and all the factors investigated. Specifically, age was indirectly related to trait anxiety ($r = -0.206$, $p < 0.001$), depression ($r = -0.193$, $p < 0.001$), anxiety ($r = -0.178$, $p < 0.001$), and stress ($r = -0.208$, $p < 0.001$). Thus, the older the students, the lower their levels of trait anxiety, depression, anxiety, and stress.

**Table 5.** Correlation between age and scale scores.

| | Age | |
|---|---|---|
| | *r* | *p*-Value |
| **STAI state** | −0.049 | 0.286 |
| **STAI trait** | −0.206 ** | <0.001 |
| **DASS depression** | −0.193 ** | <0.001 |
| **DASS anxiety** | −0.178 ** | <0.001 |
| **DASS stress** | −0.208 ** | <0.001 |

** $p < 0.001$.

Table 6 provides descriptive scores for the scales based on employment status. Student's *t*-test was performed to determine whether there were statistically significant differences between students who only attended school and those who attended school and had a paying job. There were statistically significant differences between the groups in the trait stress dimension, that is, the score for those who studied and worked (*M* = 27.25, DT = 5.54) was significantly lower than that for those who only attended school (*M* = 28.38, DT = 5.72). For the depression dimension, there were significant differences between the groups, that is, the score for students who only attended school (*M* = 5.94, DT = 4.87) was significantly higher than that for those who attended school and worked (*M* = 28.38, DT = 5.72). Similarly, there were statistically significant differences between groups in the anxiety dimension, that is, the score for students who only attended school (*M* = 4.55, DT = 4.05) was significantly higher than that for students who attended school and worked (*M* = 3.57, DT = 3.59).

**Table 6.** Descriptive and comparative scores for the scales based on employment status.

| | Employment Status, *Mean* (SD) | | Student's *t*-Test | | *d* |
|---|---|---|---|---|---|
| | **School** | **School/work** | ***t* (469)** | ***p*-Value** | |
| STAI state | 28.55 (4.42) | 28.97 (4.57) | −0.899 | 0.369 | −0.09 |
| STAI trait | 28.38 (5.72) | 27.25 (5.54) | 1.982 | 0.047 | 0.21 |
| DASS depression | 5.94 (4.87) | 4.43 (4.76) | 3.006 | 0.003 | 0.31 |
| DASS anxiety | 4.55 (4.05) | 3.57 (3.59) | 2.385 | 0.017 | 0.25 |
| DASS stress | 7.77 (4.35) | 6.95 (4.23) | 1.828 | 0.068 | 0.19 |

SD: Standard deviation; *d*: Cohen's effect size.

Table 7 provides the descriptive scores for the scales based on the University of Origin, as well as the results of Student's *t*-tests performed to determine whether there were statistically significant differences between students from the University of Granada and the University of Costa Rica. There were statistically significant differences between the groups in the trait stress dimension, that is, the score for University of Costa Rica students (*M* = 28.95, DT = 5.69) was higher than that for University of Granada students (*M* = 27.63, DT = 5.63). There were also statistically significant differences between the groups in the anxiety dimension, that is, the score for University of Costa Rica students (*M* = 4.87, DT = 4.31) was higher than that for University of Granada students (*M* = 3.98, DT = 3.72).

**Table 7.** Descriptive and comparative scores for the scales based on university.

| | University, *Mean* (SD) | | Student's *t*-Test | | *d* |
|---|---|---|---|---|---|
| | Granada | Costa Rica | *t* (469) | *p*-Value | |
| STAI state | 28.50 (4.34) | 28.98 (4.68) | −1.105 | 0.269 | −0.11 |
| STAI trait | 27.63 (5.63) | 28.95 (5.69) | −2.404 | 0.016 | −0.23 |
| DASS depression | 5.30 (4.77) | 6.00 (5.06) | −1.484 | 0.138 | −0.14 |
| DASS anxiety | 3.98 (3.72) | 4.87 (4.1) | −2.344 | 0.019 | −0.23 |
| DASS stress | 7.43 (4.24) | 7.79 (4.50) | −0.856 | 0.392 | −0.08 |

SD: Standard deviation; *d*: Cohen's effect size.

Table 8 provides the scores for the scales based on pet ownership, as well as the results of Student's *t*-tests performed to determine if there were statistically significant differences between the groups. There were no statistically significant differences between students who did or did not own a pet.

**Table 8.** Descriptive and comparative scores for the scales based on pet ownership.

| | Pet, *Mean* (SD) | | Student's *t*-Test | | *d* |
|---|---|---|---|---|---|
| | No | Yes | *t* (469) | *p*-Value | |
| STAI state | 28.12 (4.32) | 28.91 (4.51) | −1.785 | 0.075 | −0.18 |
| STAI trait | 27.80 (5.36) | 28.21 (5.83) | −0.714 | 0.475 | −0.07 |
| DASS depression | 5.62 (4.93) | 5.50 (4.86) | 0.254 | 0.8 | 0.03 |
| DASS anxiety | 4.31 (3.91) | 4.28 (3.98) | 0.09 | 0.928 | 0.01 |
| DASS stress | 8.05 (4.39) | 7.33 (4.29) | 1.7 | 0.09 | 0.17 |

SD: Standard deviation; *d*: Cohen's effect size.

To determine the possible effect of the demographic variables on STAI scores, as well as on depression and anxiety, multivariate linear regression models were employed; the results are shown in Table 9.

The model for depression was statistically significant (F(8; 932) = 22.82, *p* < 0.001), explaining 27.1% of the variability in depression. Regarding the demographic variables, none showed a statistically significant effect on depression. The STAI scale showed a significant effect: State anxiety was negative (high levels of state anxiety were associated with low levels of depression), while trait anxiety was positive (high levels of trait anxiety were associated with low levels of depression).

With respect to anxiety, the model for depression was statistically significant (F(8; 932) = 18.1, *p* < 0.001), explaining 22.6% of the variability in anxiety. With respect to demographic variables, gender showed a statistically significant effect (women had greater anxiety than men). The STAI showed a significant and direct effect on trait anxiety (high levels of trait anxiety were associated with high levels of anxiety). The assumptions of normality, independence, and homoscedasticity were verified in both models after analysis of the residuals.

**Table 9.** Effect of demographic variables and the STAI on the depression and anxiety dimensions of the DASS-21.

| | Depression | | | Anxiety | | |
|---|---|---|---|---|---|---|
| | **B (SE)** | **Beta** | *t* | **B (SE)** | **Beta** | *t* |
| Age | −0.03 (0.04) | −0.04 | −0.75 | −0.04 (0.03) | −0.08 | −1.39 |
| Gender (women vs. men) | −0.84 (0.51) | −0.07 | −1.63 | 0.99 (0.43) | 0.1 | 2.30 * |
| Relationship status (single vs. other) | 0.93 (0.66) | 0.07 | 1.41 | −0.18 (0.55) | −0.02 | −0.32 |
| Situation (school/work vs. study) | −0.37 (0.51) | −0.03 | −0.74 | −0.30 (0.42) | −0.03 | −0.70 |
| University (Granada vs. foreign) | −0.07 (0.41) | −0.01 | −0.18 | −0.50 (0.34) | −0.06 | −1.46 |
| Pet | 0.13 (0.42) | 0.01 | 0.30 | −0.18 (0.35) | −0.02 | −0.51 |
| STAI state | −0.26 (0.05) | −0.24 | −5.78 *** | −0.05 (0.04) | −0.06 | −1.31 |
| STAI trait | 0.44 (0.04) | 0.51 | 12.09 *** | 0.31 (0.03) | 0.45 | 10.32 *** |
| Assumptions | | | | | | |
| Normality [†] | | | $p = 0.211$ | | | $p = 0.200$ |
| Independence [‡] | | | 2.012 | | | 1.99 |
| Homoscedasticity [+] | | | $p = 0.579$ | | | $p = 0.411$ |

B: Non-standardized regression coefficient; SE: standard error; Beta: standardized regression coefficient; * $p < 0.05$ *** $p < 0.001$; [†] Kolmogorov–Smirnov normality test of residuals; [‡] Durbin–Whatson test; [+] Levene's test between residual and predicted values.

## 4. Discussion

Regarding age, at the international level, our results corroborate many other studies that indicate that stress and anxiety decrease as age increases, as shown by the afore-mentioned studies of Carstensen and DeLiema [25] and Valiente [26], along with those of other authors, such as Jiang [37], who addressed the moderating role of age in perceived stress during the pandemic in China. However, at the national level, studies, such as Campbell [38] and García-Ros [39] report lower levels of stress and anxiety at younger ages.

In terms of marital status, single participants showed a significantly positive difference (i.e., they had higher levels of stress and anxiety) compared to participants with other marital statuses, which also corroborates the findings of previous studies. For several decades, single status has been proposed as being associated with a greater risk of mental illness, due to its association with a greater number of stressors [40]. Subsequent studies, such as a recent study with a large sample conducted in the United States, continue to provide evidence that single people experience a higher level of perceived stress than married people [41].

Regarding sex, significant differences in the anxiety levels of men and women were evident, with women showing greater anxiety. Our data are consistent with various studies carried out in different countries, such as a recent longitudinal study of students from 11 Chinese universities [1]; the work of Schokkenbroek [42] in Belgium, which was conducted in social networks of university students; Barrera [43] with Chilean students; the study in Costa Rica conducted by Belhumeur [29]. It is worth mentioning that the results of the study by Campbell [38] are consistent with the data that we obtained; however, these researchers revealed that older men reported less stress than younger men, whereas older women reported greater stress. At the national level, our study agrees with that of García [39], who also found higher levels of stress in women. Sandín and Chorot [44] obtained similar results for students at online universities. García [39] indicated that some of these differences may be due to different patterns of socialization. One of the main debates in the academic field is based around the types of coping strategies developed by men and women [45]. From this perspective, studies, such as that of Cabanach [45], found evidence that men tend to resort to strategies of positive re-evaluation and planning to cope with problematic academic situations, whereas women essentially use social strategies, such as seeking support.

In relation to age, an indirect relationship, statistically significant was obtained with all the variables investigated. In general, it is observed that the older the student, the lower

the levels of anxiety trait, depression, anxiety and stress. Our results are consistent with those of [46] who indicate that as people mature, coping skills develop more easily and they are able to use coping strategies more successfully. We also agree with the approach of [47] in considering that, as people mature, they are more able to adopt a behavioural, cognitive and emotional strategy to cope with stressful events as they arise. Therefore, stress becomes an adaptive and psychological process, so that, as age increases, students value academic demands as stressful. We can thus affirm that, the older the students are, the better they manage stress, anxiety and depression in situations of academic overload [48], the methodological deficiencies of the teaching staff [49] or academic commitment [50]. In this way, it is clear that an adequate coping of the variables included in our research can reduce factors such as stress and improve the psychological well-being of the student, affecting the student's academic performance [51]. Following Lazarus' [52] theory of psychological stress, student coping could have two phases: The first would consist of the student's assessment and coping with what is happening, oriented towards his or her own well-being; and the second phase, which would focus on the development of efforts to manage certain external demands. From this point of view, it is evident that, depending on the assessment of the academic demand, the student will develop a type of response that could be based on his or her experience in managing stressful situations.

The results of the study regarding the work situations of students and their relationship with stress, anxiety, and depression are also consistent with those of other studies, such as Nonis and Hudson [53], and Watanabe [54], in the sense that stress, depression, and anxiety levels are lower in students who combine their studies with some type of work than in those who only study. This could be related to the influence of financial situations on stress, as we indicated previously [29]. However, in different geographical contexts, situations arise, which may differ according to labor and social conditions [55]. In the context of the present crisis, it is important to highlight the significance of some of the obtained results. Students who combine their studies and work have different reasons for doing so. There are students who work to support their family or pay for their studies and living expenses, while other students work for reasons related to lifestyle issues and/or experience, even though their relatives cover their expenses [56]. Regarding the data obtained, it is possible to say that the development of professional skills strengthens and develops academic skills.

Perhaps the most surprising aspect of our study is the lack of a significant relationship between stress levels and pet ownership, which has shown a negative correlation in existing academic and popular studies [16,57,58]. The relationship between stress levels and the presence of domestic animals, especially dogs, has been shown in many studies on the basis of both individuals' perceptions and changes in physiological variables [31,32,59,60]. Based on our results, we highlight that in future research, it would be interesting to compare perceived stress and stress responses as measured by cortisol levels. Such research could allow variables that influence stress in students who have pets and those who do not to be specified in a more comprehensive manner. This consideration leads us to think that it would be interesting to conduct a more in-depth study on the development of a canine therapy program to reduce stress during examination periods. We recognize that this would entail a different approach, starting with the conception that higher levels of stress are usually present during examination periods [61]; therefore, interaction with a dog could allow a more exhaustive analysis of how stress is regulated at specific times [16] and not only during long periods, without specifying the precise moment when stress might occur. This proposal is based on the premise that improving students' well-being has been included in university policy agendas at the international level.

We obtained several significant results using the STAI scale. On the one hand, state anxiety was negative; thus, high levels of state anxiety in students are associated with low levels of depression. On the other hand, trait anxiety was positive, and high levels of trait anxiety are associated with high levels of depression. In this way, although students have a high level of state anxiety at specific times and present some intensity of processes or empirical reactions, it cannot be concluded that this anxiety leads to a significant decline in

social, interpersonal, or occupational functioning, as is characteristic of depression. However, the propensity toward anxiety, which may cause individuals to perceive situations as threatening and, as a result, experience greater anxiety, is associated with increased levels of depression, which can affect social, interpersonal, and occupational functioning. In this way, we agree with the studies of Ninan and Berger [14] that anxiety disorders in students can be regarded as a risk indicator for the onset of depression.

**Author Contributions:** Conceptualization, E.J.L.S. and M.K.G.; methodology, M.K.G.; software, M.K.G.; validation, J.G.P. and M.C.G.M.; formal analysis E.J.L.S. and J.G.P. All authors have read and agreed to the published version of the manuscript.

**Funding:** This research was funded by Funded by Vice-Rectorat for Outreach and Heritage & Faculty of Education, University of Granada. Programme, 2020 Programme of Activities of Outreach. Project: Stress-Less: "Take My Paws". Fostering Student Wellness with a Therapy Dog Program at the Faculty Library.

**Institutional Review Board Statement:** Authorization was obtained from the ethics committee of the University of Granada (nº: 1650/CEIH/2020).

**Informed Consent Statement:** Informed consent was obtained from all subjects involved in the study.

**Data Availability Statement:** Data of this study are available upon reasonable request from the corresponding author.

**Conflicts of Interest:** The authors declare no conflict of interest.

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
