# Peer review of "Levels of Stress, Anxiety, and Depression in University Students from Spain and Costa Rica during Periods of Confinement and Virtual Learning"

_education, doi:10.3390/educsci12100660_

Round 1

Reviewer 1 Report

This is an interesting paper on one of the most studied aspects for years in relation to teaching, the problem of anxiety and stress in students. On this occasion, the authors investigate the stress and anxiety caused by students at two universities during the first months of the COVID pandemic and the consequence of virtual teaching.

The methodology and procedures are correct for the objectives sought and, at the same time, sufficient references from previous works are provided for a correct discussion of the results obtained. However, there are some content and format issues that need to be reviewed and corrected:

- In line 43 the word "concentration" is not clear. Do they refer to concentration problems? in that case it would be more appropriate to use "concentration and memory problems"

- Lines 110 to 112 and 122 to 124 are exactly the same. If you want to leave both (I don't think it's necessary), it would be more correct to slightly modify one of them.

- In table 1, the last two rows are not properly aligned under their column and are shifted to the left.

- Could you explain how the questionnaires were carried out? through a questionnaire in Word delivered that is filled out and sent, through an online questionnaire.…

- Is it possible to discuss a little more what could be the reason for the difference observed according to age? perhaps due to better management of new technologies or greater maturity and better organization of time... it is necessary to discuss these results a little more, as has been done with other factors such as gender or marital status…

- The differences observed between students from Spain and Costa Rica should be further explained and discussed.

- References 34 to 36 do not follow exactly the same format as the rest of the references. The rules on reference formats should be carefully reviewed.

Author Response

Dear Reviewer We have carried out all the instructions. We attach the article with the modifications. kind regards

Reviewer 2 Report

The title as well as the introduction raised expectations about your manuscript and research. The topic you are addressing would be a relevant addition to existing literature. Thank you for this valuable contribution. I will structure my feedback in (a) general remarks (these comments cover feedback applicable in the entire manuscript), and (b) specific remarks (feedback on sentence and/or word level). The specific remarks can include a quote from your original manuscript to refer to a specific section. The specific remarks will refer to page (emphasis added in boldface; e.g., 1.15/16) and row(s; e.g., 11.15/16).

General remarks:

The overall manuscript is neat and written concisely—with relevant information for existing literature. One aspect that you can focus on is consistency in terminology and punctuation. Early on in your manuscript, there are odd verbs used to describe the phases in the research process. Read existing literature to familiarize yourself with common (or frequently used) verbs. I would also suggest to let a native English speaker read over your manuscript.  

Specific remarks:

p.1.5/6             “more prevalent” à than what? Compared to what?

p.1.6                Avoid the intensifier “very”. Improve your argument. These intensifiers are only necessary if you have an invalid argument. If you argument is sufficient, these intensifiers become redundant. Moreover, I am wondering what the addition of “in very different geographical areas” adds to your message.

p.1.10              The concept “virtual learning” requires attention in your theoretical framework.

p.1.8/11           The pandemic will be one of your main concepts. Therefore, you need to write it consistently throughout your manuscript.

p.1.12              The word “the” is redundant (in “both the universities”). Furthermore, you do not apply questionnaires; you administer questionnaires.

p.1.15              The “d” in Cohen’s d needs to be placed in italics. I was wondering about your arguments to use Cohen’s d instead of Hedges’ g; however, your sample size can be labeled as large.

p.1.19              You need to add the practical implications. Will it be relevant for policy makers? Educators?

p.1.20              I would also add “mental health” as a keyword.

p.1.23              I already mentioned that the terms about the COVID-19 pandemic need to be consistent throughout your manuscript. Thus, you keep using COVID-19 pandemic, or you mention here that from this point onward you will only use COVID-19.

p.1.32              Avoid back-to-back brackets: (x) [y].

p.1.38              You need to end the sentence with a period.

p.1.32–38        What is the goal of this paragraph?

p.1.39              I am also wondering what the point is you are trying to make. I would suggest to restructure this section because there is a lack of coherence.

p.1.40              I would place (a), (b), and (c) before the indicator.

p.2.47–63        Try to aim for more coherence. Starting two paragraphs with “According to” is not the way to do that. Moreover, the last sentence of your introduction should be linked to the goal of your study. Also the remainder of your theoretical framework (e.g., section 1.1), I notice a lack of coherence. Go over what the different sections need to contain (Introduction and Theoretical framework).

p.2.75–81        You present two short paragraphs, but both aren’t paragraphs. A paragraph is at least three sentences. Merge them to a meaningful whole or add information. The section is fragmented. Also the first paragraphs on page 3 aren’t paragraphs.

p.2.77/94         Sometimes you use a comma to separate two references and sometimes you use a semi colon. Use a comma and apply that consistently.

p.Table 1        Make a note that you will present the percentages with one decimal numbers as opposed to the n.

p.3.128            You can merge this with the sentence before (it is an extension of that sentence).

p.4.131            3-point Likert scale.

p.4.137            Avoid back-to-back brackets.

p.4.137–141    What type of scale do you use to answer these items?

p.4.143            You do not apply questionnaires. You administer them. Please revise the verb (also in rows 146/147).

p.4.159            The “r” needs to be placed in italics. In a similar vein, the “d” needs to be in italics and the p in row 166. Make sure you check the remainder of your manuscript for this. This also applies to the M and

p.section 2.2   Can you also describe the administration setting?

p.4.169/170     This has to be introduced in the methodology. Moreover, my comment about the decimals numbers applies here as well.

p.5.200            Use comma’s and periods consistently. Compare: 4.51 and 4,01 (row 200).

Author Response

Dear Reviewer We have carried out all the instructions. We attach the article with the modifications. We also answer the specific questions that the reviewer asked us kind regards.

p.1.5/6             “more prevalent” à than what? Compared to what?

Mental health problems, specifically those related to stress, anxiety and depression have become more prevalent among university students prior to COVID levels.

p.1.6                Avoid the intensifier “very”. Improve your argument. These intensifiers are only necessary if you have an invalid argument. If you argument is sufficient, these intensifiers become redundant. Moreover, I am wondering what the addition of “in very different geographical areas” adds to your message.

With that sentence we want to comment that regardless of the geographical area, there is a high prevalence of symptoms of depression and anxiety in university students compared to pre-pandemic levels. Therefore, in general, we can indicate that there was an increase in symptoms of depression and anxiety at the university level during the COVID pandemic.

p.1.10              The concept “virtual learning” requires attention in your theoretical framework.

Dear editor, the authors do not understand this indication. We have put the consistency of the use of this concept in the Discussion section. Our study is not based on the influence of e-learning, but on coping with various variables that influence levels of stress, anxiety and depression. All of this is included in the Discussion section.

p.1.8/11           The pandemic will be one of your main concepts. Therefore, you need to write it consistently throughout your manuscript.

The conception and influence of pandemics is described throughout the article. In all sections we refer to the term pandemic implicitly. As it is an ambiguous indication, we do not know what specific changes the reviewer wants us to make.
